**Data Availability Statement:** All relevant data are within the paper and its Supporting Information files.

**Funding:** JYH received a grant from the National Research Foundation of Korea (NRF) funded by the

# Rapid drug increase and early onset of levodopa-induced dyskinesia in Parkinson's disease

**Jin Yong Hong**[1], **Mun Kyung Sunwoo**[2], **Jung Han Yoon**[3], **Suk Yun Kang**[4], **Young H. Sohn**[5], **Phil Hyu Lee**[5,6]☯*, **Seo Hyun Kim**[1]☯*

**1** Department of Neurology, Yonsei University Wonju College of Medicine, Wonju, Gangwon-do, South Korea, **2** Department of Neurology, Bundang Jesaeng General Hospital, Seongnam, Gyeonggi-do, South Korea, **3** Department of Neurology, Ajou University School of Medicine, Suwon, Gyeonggi-do, South Korea, **4** Department of Neurology, Dongtan Sacred Heart Hospital, Hallym University College of Medicine, Hwaseong, Gyeonggi-do, South Korea, **5** Department of Neurology and Brain Research Institute, Yonsei University College of Medicine, Seoul, South Korea, **6** Severance Biomedical Science Institute, Yonsei University College of Medicine, Seoul, South Korea

☯ These authors contributed equally to this work.
* phlee@yuhs.ac (PHL); s-hkim@yonsei.ac.kr (SHK)

## Abstract

A higher levodopa dose is a strong risk factor for levodopa-induced dyskinesia (LID) in patients with Parkinson's disease (PD). However, levodopa dose can change during long-term medication. We explored the relationship between levodopa dose and time to onset of LID using longitudinal multicenter data. Medical records of 150 patients who were diagnosed with de novo PD and treated with levodopa until onset of LID were collected. Levodopa dose were assessed as the dose at 6 months from levodopa initiation and rate of dose increase between 6 months and onset of LID. The groups with earlier LID onset had higher levodopa and levodopa-equivalent dose at the first 6 months of treatment and rapid increase in both levodopa and levodopa-equivalent dose. Multivariable linear regression models revealed that female sex, severe motor symptom at levodopa initiation, and higher rate of increase in both levodopa and levodopa-equivalent dose were significantly associated with early onset of LID. The present results demonstrated that rapid increase in levodopa dose or levodopa-equivalent dose is associated with early onset of LID.

## Introduction

Levodopa improves motor symptoms in patients with Parkinson's disease (PD). However, many patients treated with levodopa chronically encounter levodopa-induced dyskinesia (LID). LID develops in approximately 60% of patients within at 5 years of levodopa therapy [1–3], and it eventually affects most of patients treated with levodopa [1, 4, 5].

Previous studies have determined risk factors for LID to be younger age at PD onset [1, 4, 6, 7], female sex [7], lower body weight [7], higher levodopa dose [1, 4, 6, 7], and more severe motor symptoms at baseline [7–9], lower dopamine transporter availability at baseline [6, 9],

Korea government (MSIT) (grant number 2017R1C1B5076522). The funder had no role in study design, data collection and analysis, decision to publish, or preparation of the manuscript.

**Competing interests:** The authors have declared that no competing interests exist.

and higher dopamine turnover rate in the posterior putamen at baseline [10]. Among these risk factors, high levodopa dose is one of the most commonly reported; however, there was no established method for assessing levodopa dose which is usually increased during long-term therapy. In previous studies, levodopa dose has been defined as the initial dose [1, 4], cumulative dose [8], or dose at LID onset/last dose in LID-free patients [6, 7]. However, the levodopa dose usually increases with disease progression in general practice and clinical trials [3, 11], and the rate of dose increase varies among patients.

Results from clinical trials consistently showed that parkinsonian motor symptoms improved within 6 to 9 months from the initiation of medication and then progressed again with a constant rate [2, 3, 12, 13]. Similar to clinical trials, observational studies have also shown that the initial drug dose is titrated within 6 to 12 months from the initiation of medication [1, 4, 14], and the rate of subsequent dose increases is similar for many years [14]. Therefore, assessing levodopa dose using initial titration dose and rate of dose increase can help analyzing the longitudinal effect of levodopa dose on development of LID.

In this study, we collected longitudinal data on levodopa dose from patients with PD who experienced LID during medical therapy, and we explored the relationship between levodopa dose and time to development of LID.

## Materials and methods

### Subjects

We reviewed medical records from 4 referral hospitals in South Korea (Wonju Severance Christian Hospital, Bundang Jesaeng General Hospital, Severance Hospital, Dongtan Sacred Heart Hospital, and Severance Hospital) between March 2007 and October 2018 and selected data of PD patients who experienced LID. PD was diagnosed according to the clinical criteria of UK PD brain bank [15], and patients who showed atypical features listed in step 2 of the criteria during treatment period were excluded from the study. Patients who had been treated with dopaminergic drugs or dopamine receptor blocking agents before visiting our hospitals were also excluded. Among them, we selected patients who initiated medical treatment with levodopa. Patients who took dopamine agonist or monoamine oxidase B (MAO-B) inhibitor prior to levodopa initiation, a dopamine-receptor blocking agent during treatment period, or amantadine before development of LID were also excluded. Patients with impulse control disorders, psychosis, or dementia were not included because these conditions may affect drug dose.

LID was assessed based on medical record. For regular management for PD, patients usually have been visiting outpatient clinic of 5 experts in movement disorders (J.Y.H., M.K.S., S.Y.K., Y.H.S., and P.H.L.) every 2–3 months. When patient reported symptom suggesting LID or dyskinetic movement was observed in the clinic, clinicians recorded the information for LID. The onset of LID was defined as 1) the first day that LID was observed by clinician when patient did not perceive their dyskinesia, 2) the day indicated by patient when LID was observed by clinician and the patient remembered the day that LID had begun. Patients who skipped visit more than 6 months were excluded from the study. Because short latency to onset of LID may make rate of dose increase overestimated, patients who LID developed within 2 years from levodopa initiation were excluded.

Subjects were grouped according to time to development of LID from initiation of levodopa: E (Early, 2 years < time to LID onset ≤ 4 years), M (Middle, 4 years < time to LID onset ≤ 6 years), L (Late, 6 years < time to LID onset).

Motor symptoms were assessed using Unified PD Rating Scale (UPDRS) motor score (part III) and modified Hoehn and Yahr stage, and the scores rated at *de novo* status were used in this study.

### Ethics statement

The study protocol was approved by the Institutional Review Board (IRB) of the Wonju Severance Christian Hospital (approval number: 2018-10-0007), and the study was conducted according to the Declaration of Helsinki. The requirement for written informed consent was exempted by the IRB and local regulation due to the retrospective design.

### Drug dose

Patients took equal dose of levodopa three times a day, and the treatment was usually initiated with 50mg of levodopa three times a day. When dose increase was needed, the single dose was usually increased by 50mg (150mg a day). Data for the dose of antiparkinsonian drugs at 6 months, 1 year, and every year after that from initiation of levodopa were collected from prescription records. The levodopa-equivalent dose (LED) was calculated according to a published formulation [16].

To reflect change of drug dose over disease progression, the drug dose was assessed as initial titration dose and rate of dose increase. Initial titration dose was defined as the daily dose of levodopa at 6 months from the initiation of medication [14], and the rate of dose increase was calculated as following: {(drug dose at onset of LID)–(drug dose at 6 months [initial titration dose])} / {(years at onset of LID)–(0.5 year)}. Initial titration dose and rate of dose increase for LED were calculated with same method.

### Statistical analyses

Analysis of variance (ANOVA) and chi-square test were used to compare groups, and *post hoc* analyses were conducted with Bonferroni's method. Factors associated with time to LID onset were explored using Pearson's correlation and multivariable linear regression models with stepwise selection method. Because data for UPDRS motor score of 61 patients were missed, linear regression analyses were conducted with data from 89 patients. Statistical analyses were performed with SPSS Statistics 23 (IBM SPSS, Armonk, NY, USA), and $p < 0.05$ was considered significant. Graphic illustrations were obtained using GraphPad Prism version 7.02 for Windows (GraphPad Software, La Jolla, CA, USA).

## Results

### Demographic and clinical characteristics

According to the inclusion and exclusion criteria of this study, a total of 150 patients were collected for this study (58 for E, 51 for M, and 41 for L group). Demographic and clinical characteristics of study subjects are presented in Table 1. Sex, age at PD onset, age at initiation of levodopa therapy, and time between PD onset and initiation of levodopa therapy did not differ significantly among groups. UPDRS motor score at levodopa initiation were higher in E (29.2 vs. 22.1, *post hoc p* = 0.029) than L group, however modified Hoehn and Yahr stage was not different among groups.

### Drug dose

The mean levodopa dose and LED over time according to the groups are shown in Fig 1, and data for initial titration dose and rate of dose increase are presented in Table 1. Initial titration

**Table 1. Demographic and clinical characteristics of subjects and drug dose.**

| | Total (n = 150) | E (n = 58) | M (n = 51) | L (n = 41) | p-value | Intergroup comparison |
|---|---|---|---|---|---|---|
| Female, n (%) | 94 (62.7) | 40 (69.0) | 29 (56.9) | 25 (61.0) | 0.413 | |
| Age at PD onset, year | 63.9 ± 8.7 | 62.9 ± 8.9 | 64.0 ± 8.4 | 65.1 ± 8.6 | 0.469 | |
| Age at levodopa initiation, year | 65.3 ± 8.4 | 64.4 ± 8.6 | 65.4 ± 8.2 | 66.5 ± 8.6 | 0.470 | |
| PD onset to levodopa initiation, year | 1.4 ± 1.2 | 1.5 ± 1.2 | 1.4 ± 1.1 | 1.4 ± 1.4 | 0.842 | |
| UPDRS motor score [a] | 27.2 ± 10.0 | 29.2 ± 8.5 | 27.7 ± 11.9 | 22.1 ± 8.5 | 0.033 | E>L |
| Modified Hoehn & Yahr stage [b] | 2.2 ± 0.5 | 2.2 ± 0.5 | 2.1 ± 0.5 | 2.0 ± 0.3 | 0.283 | |
| Levodopa dose | | | | | | |
| Initial titration dose (at 6 months) | 383.3 ± 140.9 | 406.9 ± 161.1 | 382.6 ± 133.2 | 350.6 ± 113.5 | 0.147 | |
| Rate of increase, mg/day/year | 54.1 ± 55.5 | 69.9 ± 70.6 | 46.1 ± 45.7 | 41.5 ± 33.9 | 0.019 | E>L |
| Dose at LID onset, mg/day | 596.7 ± 220.6 | 578.0 ± 226.0 | 583.8 ± 190.1 | 639.0 ± 246.5 | 0.353 | |
| Levodopa-equivalent dose | | | | | | |
| Initial titration dose (at 6 months) | 505.5 ± 154.7 | 518.6 ± 168.9 | 517.8 ± 148.4 | 460.9 ± 135.9 | 0.129 | |
| Rate of increase, mg/day/year | 87.1 ± 74.2 | 113.8 ± 91.6 | 75.8 ± 63.5 | 63.3 ± 41.2 | 0.001 | E>M, E>L |
| Dose at LID onset, mg/day | 841.3 ± 258.4 | 797.6 ± 262.1 | 848.3 ± 253.4 | 894.4 ± 254.7 | 0.181 | |
| Drug use at LID onset, n (%) | | | | | | |
| Dopamine agonists | 119 (79.3) | 41 (70.7) | 45 (88.2) | 33 (80.5) | 0.076 | |
| MAO-B inhibitors | 44 (29.3) | 14 (24.1) | 17 (33.3) | 13 (31.7) | 0.532 | |
| Entacapone | 60 (40.0) | 20 (34.5) | 19 (37.3) | 21 (51.2) | 0.218 | |
| Anticholinergics | 19 (12.7) | 8 (13.8) | 7 (13.7) | 4 (9.8) | 0.806 | |

PD, Parkinson's disease; UPDRS, Unified PD rating scale; LID, levodopa-induced dyskinesia; MAO-B, monoamine oxidase B.

[a] Data of 89 patients (41 for E, 29 for M, and 19 for L group) existed.

[b] Data of 92 patients (42 for E, 29 for M, and 21 for L group) existed.

dose of levodopa and LED were similar among groups. The rate of levodopa increase was higher in E group than L group (69.9 vs. 41.5 mg/day/year, *post hoc p* = 0.034), and rate of LED increase was higher in E group than both M (113.8 vs. 75.8 mg/day/year, *post hoc p* = 0.019) and L groups (113.8 vs. 63.3 mg/day/year, *post hoc p* = 0.002). Both levodopa dose and LED at onset of LID were not significantly different among groups. The frequency of use of other anti-parkinsonian drugs other than levodopa at onset of LID did not differ among groups.

## Risk factor for early LID

Pearson's correlation analysis (Fig 2) revealed that time to onset of LID was correlated with initial titration dose of levodopa (r = -0.17, p = 0.043), rate of dose increase for levodopa (r = -0.21, p = 0.009) and LED (r = -0.29, p<0.001), UPDRS motor score (r = -0.31, p = 0.003), and modified Hoehn and Yahr stage (r = -0.22, p = 0.033). Initial titration dose of LED tended to correlate with time to onset of LID (r = -0.15, p = 0.066). However, time to onset of LID was not associated with age at onset of PD, age at levodopa initiation, time between PD onset and initiation of levodopa therapy, sex difference, and use of dopamine agonist, MAO-B inhibitor, entacapone, or anticholinergics.

Multivariable linear regression models for levodopa dose (F = 7.204, p<0.001, $R^2$ = 0.203) exhibits that female sex (β = -0.846, p = 0.028), higher rate of levodopa increase (β = -0.010, p = 0.004), and higher UPDRS motor score at baseline (β = -0.054, p = 0.003) were associated with early onset of LID (Table 2). Linear regression model for LED (F = 9.716, p<0.001, $R^2$ = 0.255) also found that female sex (β = -0.831, p = 0.024), higher rate of LED increase

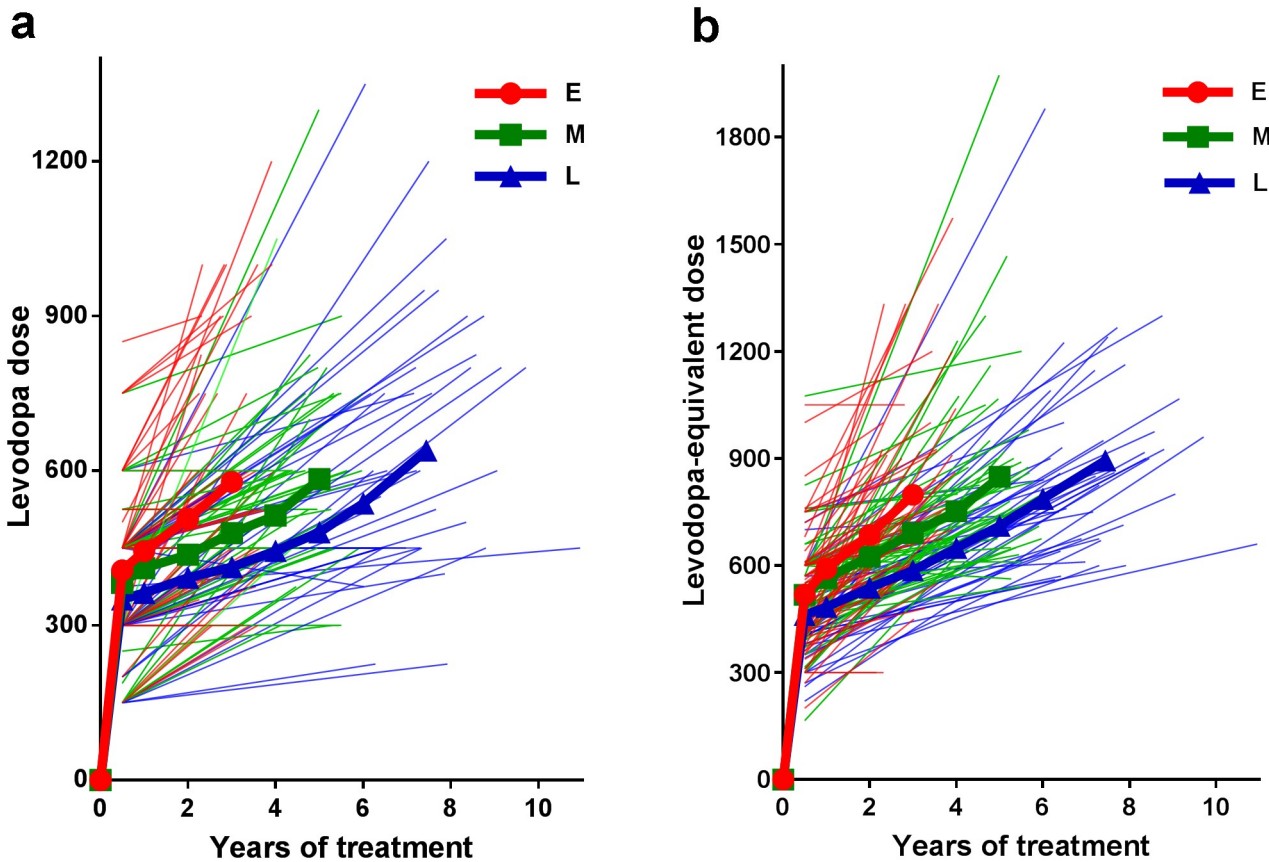

**Fig 1. Levodopa and levodopa equivalent dose.** Levodopa (a) and levodopa equivalent dose (b) of each subjects. Left and right ends of each light line indicate initial titration dose and dose at onset of LID of each subject, respectively. The mean doses of subject groups are expressed as heavy lines.

(β = -0.009, p<0.001), higher UPDRS motor score at baseline (β = -0.053, p = 0.003) were associated with early onset of LID.

## Discussion

The present study was conducted with a relatively large number of patients with LID and explored the effect of longitudinal changes in levodopa dose or LED on time to onset of LID. The results demonstrated that higher rate of dose increase of dopaminergic drugs were significantly associated with the early onset of LID.

Patients who developed LID early had more severe motor symptoms at initiation of levodopa therapy than did patients with late onset of LID, as in previous studies [7–9]. Functional neuroimaging studies also demonstrated that lower dopaminergic activity in de novo patients predicts early LID onset [6, 9, 10, 17]. The association between early onset of LID and higher initial levodopa dose was also observed in previous studies. The mean levodopa dose over the first 6 months [1] and maintenance dose of levodopa within the first year [4] were predictors of early onset of LID. Given that motor symptoms and initial dose of dopaminergic drug may reflect the severity of motor symptoms at diagnosis, these findings agree with previous reports suggesting that more severe dopaminergic denervation when beginning treatment is a risk factor for LID [1, 4]. In the present study, initial titration dose of levodopa or LED showed significant negative correlation with time to onset of LID, however the final regression models

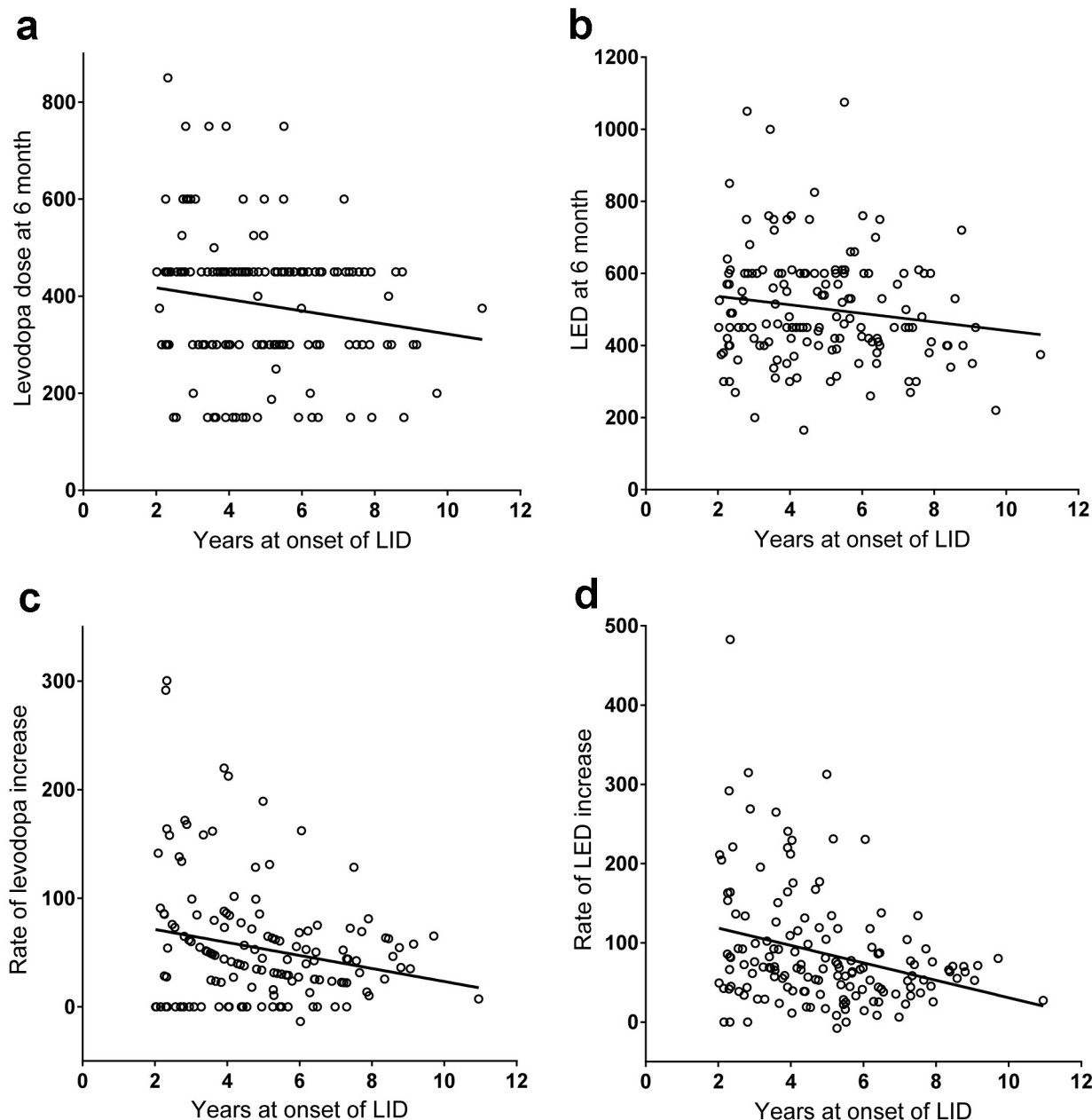

**Fig 2. Time to onset of levodopa-induced dyskinesia and drug dose.** Correlation between time to onset of levodopa-induced dyskinesia (LID) and initial titrating levodopa dose (a), time to LID and initial titrating levodopa-equivalent dose (b), time to LID and rate of increase in levodopa dose (c), and time to LID and rate of increase in levodopa-equivalent dose (d).

included UPDRS motor score at baseline instead of initial titration dose. This implies that initial titration dose does not act as an independent risk factor for early onset of LID, but just reflects the severity of motor symptom at initiation of levodopa therapy.

The present data demonstrated that the rate of increase in levodopa dose or LED during treatment is a significant predictor of early onset of LID. Generally, physicians increase the drug dose when the motor symptoms get worse, which means that the rate of dose increase may reflect disease progression. A recent study using serial SPECT imaging data also demonstrated

**Table 2. Results of multiple linear regression analyses to predict time to levodopa-induced dyskinesia.**

| Factors | Levodopa [a] | | | Levodopa-equivalent dose [b] | | |
|---|---|---|---|---|---|---|
| | β | 95% CI | p-value | β | 95% CI | p-value |
| Female | -0.846 | -1.599, -0.093 | 0.028 | -0.831 | -1.552, -0.111 | 0.024 |
| Rate of dose increase | -0.010 | -0.017, -0.003 | 0.004 | -0.009 | -0.014, -0.005 | <0.001 |
| UPDRS motor score at levodopa initiation | -0.054 | -0.090, -0.019 | 0.003 | -0.053 | -0.087, -0.018 | 0.003 |

CI, confidential interval; UPDRS, Unified Parkinson's Disease Rating Scale.

[a] $F = 7.204$, p<0.001, $R^2 = 0.203$.

[b] $F = 9.716$, p<0.001, $R^2 = 0.255$.

more rapid decrease on putaminal dopamine transporter activity in patients with LID compared with patients without LID [9]. Therefore, the present results supports that rapid degeneration of the dopaminergic system is associated with early onset LID.

The levodopa dose and LED at onset of LID of individual patient varied, however the mean dose at onset of LID among patients groups were not different. This suggests that reaching particular level of levodopa dose may be associated with onset of LID. Both higher initial titration dose and rapid dose increase shorten the time to reach a patient-specific drug threshold where the LID develops. However the present results cannot determine which is related with the early onset of LID, the severity of dopaminergic deficit or higher levodopa dose. No clinical trials with a fixed dose of levodopa for several years have been conducted, therefore it is unclear that levodopa dose is the independent risk factor for early onset of LID regardless of severity of dopaminergic denervation. On the other hand, several previous studies have shown that a single administration of overdosed levodopa also induces peak-dose dyskinesia in patients with advanced PD who have never experienced LID before [18, 19], which suggests that long-term exposure to levodopa is not a prerequisite for development of LID in some patients. Therefore, further studies with more evidence are required to clarify whether dopaminergic drug truly induces LID or simply acts as a trigger for onset.

Several limitations of this study need to be addressed. Drawing concrete conclusions from this study is difficult because of its retrospective design without randomized allocation of drug use. Many factors can influence physician choice of drug dose, such as patient age, drug compliance, cognitive status, psychiatric symptoms, and adverse events. Finally, patients who experienced LID within 2 years from levodopa initiation were excluded due to the study design, therefore it is unclear that the study findings are valid in patients with very early LID.

In conclusion, present study demonstrated that rapid dose increase of dopaminergic drug is associated with early onset of LID. However, further studies are required to reveal which is the determinant of onset of LID, severity of the disease or dose of dopaminergic drug.

## Author Contributions

**Conceptualization:** Jin Yong Hong, Phil Hyu Lee, Seo Hyun Kim.

**Data curation:** Jin Yong Hong, Mun Kyung Sunwoo, Jung Han Yoon, Suk Yun Kang, Young H. Sohn, Phil Hyu Lee.

**Formal analysis:** Jin Yong Hong.

**Funding acquisition:** Jin Yong Hong.

**Investigation:** Jin Yong Hong.

**Supervision:** Phil Hyu Lee, Seo Hyun Kim.

**Visualization:** Jin Yong Hong.

**Writing – original draft:** Jin Yong Hong.

**Writing – review & editing:** Mun Kyung Sunwoo, Jung Han Yoon, Suk Yun Kang, Young H. Sohn, Phil Hyu Lee, Seo Hyun Kim.

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
