## [Decision Letter · Decision Letter 0]

23 Jun 2020

PONE-D-20-13495

Rapid drug increase and early onset of levodopa-induced dyskinesia in Parkinson's disease

PLOS ONE

Dear Dr. Seo Hyun Kim,

I thank you for submitting your manuscript to PLOS ONE. After careful consideration, we feel that it has merit but does not fully meet PLOS ONE’s publication criteria as it currently stands. Therefore, we invite you to submit a revised version of the manuscript that addresses the points raised during the review process.

More presicely, the authors should add more details regarding the methods and the results sections. The paper would also be improved if the discussion could be expanded.

We look forward to receiving your revised manuscript.

Kind regards,

Véronique Sgambato

Academic Editor

PLOS ONE

Journal Requirements:

2. In your Methods section, please provide additional information about the participant recruitment method and the demographic details of your participants.

Please ensure you have provided sufficient details to replicate the analyses such as: a) a description of how records were chosen, and b) descriptions of where the research took place.

Reviewers' comments:

Reviewer's Responses to Questions

**Comments to the Author**

1. Is the manuscript technically sound, and do the data support the conclusions?

Reviewer #1: Yes

Reviewer #2: Yes

2. Has the statistical analysis been performed appropriately and rigorously? 

Reviewer #1: Yes

Reviewer #2: Yes

3. Have the authors made all data underlying the findings in their manuscript fully available?

Reviewer #1: Yes

Reviewer #2: Yes

4. Is the manuscript presented in an intelligible fashion and written in standard English?

Reviewer #1: Yes

Reviewer #2: Yes

5. Review Comments to the Author

Reviewer #1: Rapid drug increase and early onset of levodopa-induced dyskinesia in Parkinson's disease

PONE-D-20-13495

Research article for the journal Plos One

The authors conducted a retrospective multicenter study on the occurrence of levodopa induced dyskinesia (LID) in Parkinson Disease (PD) Patients.

They studied the relationships between the increase in levodopa dose and levodopa equivalent dose (LED) over years and the time of LID onset.

They included 150 PD patients from 4 centers and divided them in 3 subgroups, those who develop early LID (>2 years to � 4 years), those who develop middle LID (> 4 years and � 6 years) and late LID ( > 6 years).

They found that the higher rate of increase in levodopa dose and LED, the earlier the onset of LID.

They concluded that dopaminergic drug could induce LID or act as a trigger for onset.

The study addresses an important issue for the management of PD patients.

I have some comments and questions.

Could the authors precise how they included the subjects in their study: did they select patients who were known to have early/middle and late LID onset or did they included consecutive patients followed in the different centers and then, classified them in 3 different groups, a posteriori?

Could the authors argue the reason for which they excluded patients with very early LID? It would have been interesting to include them as they usually reflect severe dopamine depletion.

Which criteria did they use to consider that patients were at LID onset? Did they took the medical observation into consideration? The (MDS)-UPDRS part IV for complications of treatment? The Marconi LID scale? Patients questionnaire? We know that slight dyskinesia may occur early in disease evolution but patients are lot always aware of them and are not always able to recognize them as dyskinesia. It may be, thus, a limit to define the precise onset of LID.

In Table 1 the authors reported the number of patients with treatment other than levodopa: was it the treatment, patients had, at time of LID onset?

In the results section the authors mentioned that female subjects were at higher risk of early LID but they wrote that male subjects were at higher risk in the abstract: please correct.

The authors associated the risk of LID onset with the rapid rate of levodopa /LED increase. However, it is known that LID are also associated with pulsatile post-synaptic dopaminergic stimulation.

If data is available, and in order to help interpreting the results of the study, the authors could study and report the way levodopa was prescribed and whether the increase in levodopa dose or LED was associated with an increase in levodopa/LED unitary dose (which may enhance pulsatile answer and promote LID) or whether the increase was associated an increase in the number of daily intakes (a more continuous dopaminergic administration, which may delay the onset of LID).

If the authors could add these data, they could develop more precisely the discussion on the pathophysiology of LID and the way to manage PD patients to avoid the occurrence of LID, even in severely depleted patients.

Reviewer #2: Manuscript Number: PONE-D-20-13495

Manuscript Title: Rapid drug increase and early onset of levodopa-induced dyskinesia in Parkinson's disease

Authors assessed a retrospective sudy about the relationship between levodopa dose and time onset of Levodopa-induced dyskinesia (LID). This is an interesting paper that reinforces previous results . The article is well written and analyses are accurately performed.

However, I have some comments about the data and their interpretation.

Major comments

MATERIALS AND METHODS

Subjects :

- Authors chose to perform the analyses with de novo PD patients. It is well known that some PD patients are in fact misdiagnosed (Coarelli et al, J Neurol Sci, 2019 ; Hustad E, Aasly JO. Front Neurol. 2020). How did they ensure that these patients were truly PD patients and not parkinsonian patients from other degenerative disorders such as PSP, MSA,… Which clinical criteria did they use? Did they check these criteria at the beginning of the « PD » disease but also during the follow-up of these patients? It seems strange that authors did not observe other parkinsonian disorders due to initial misdiagnosis, even if they are experts in movement disorders.

- Did the authors use a score of dyskinesia such as the UDysRS?

- In the sentence « Data of patients with… drug dose », the « for » seems not appropriate or something is missing.

- How was performed the motor UPDRS by authors? Was it applied in the OFF state or in the ON state (or intermediate state)? How long after the last medication did the UPDRS have been carried out? Please detail the assessment of the UPDRS score.

RESULTS

- Demographic characteristics : it is probably more accurate to rename this part « Demographic and clinical characteristics »

- The UPDRS motor score was available for only 89 out of 150 patients. Could you precise how many UPDRS motor scores were available in each group (E, M, L). Moreover, in which condition (On? OFF? intermediate?) were assessed these scores?

- The UPDRS is relatively elevated at initiation of Levodopa and/or dopaminergic agonists. It seems that patients were not sufficiently treated. Could you please justify this point?

- Entacapone increases the occurrence of dyskinesia (Ahn et al., J Clin Neurol. 2007 ; Trenkwalder et al., Neurology. 2019). Did authors observe any difference in dyskinesia occurrence between patients who took entacapone and those who did not take it? Did they look for a difference in each group (E, M, L)?

- In the « Risk factor for early LID » section, second line, authors declared that there was a correlation between time to onset of LID and LED but the p value is 0.066 which is above 0.05… (you should also correct the sentence related with this result in the Discussion section)

DISCUSSION

Concerning the sentences from «However the present results… » to « … acts as a trigger for onset » :

It is well established that the occurrence of LID is secondary both to the dopaminergic denervation AND the levodopa administration. Without one of this factor, LID could not appear. LID occur in patients (or animals) taking Levodopa but only in the presence of dopaminergic degeneration.

Minor comments

- Replace « Table 2 » by « Table 1 » in the « Drug » section of the Results.

- Insert « Table 2 » at the end of the sentence « Multivariable linear regression models … were associated with early onset of LID ». Same remark for the next sentence.

6. PLOS authors have the option to publish the peer review history of their article (what does this mean?). If published, this will include your full peer review and any attached files.

Reviewer #1: No

Reviewer #2: No

---

## [Author Response · Author response to Decision Letter 0]

2 Jul 2020

Reviewer #1

- Could the authors precise how they included the subjects in their study: did they select patients who were known to have early/middle and late LID onset or did they included consecutive patients followed in the different centers and then, classified them in 3 different groups, a posteriori?

Answer) This study was conducted using medical record and study subjects were retrospectively selected according to the study criteria. Now we clarify and rewrite the Method section in revised manuscript.

- Could the authors argue the reason for which they excluded patients with very early LID? It would have been interesting to include them as they usually reflect severe dopamine depletion.

Answer) Importance of this study is on the method for assessment of drug dose. The rate of dose increase was calculated as (change in dose / treatment duration), therefore relatively short duration of treatment may make the rate of dose increase overestimated. To reduce this chance, we selected patients who encountered LID at least two years after initiation of levodopa. We declare this in Method section.

- Which criteria did they use to consider that patients were at LID onset? Did they took the medical observation into consideration? The (MDS)-UPDRS part IV for complications of treatment? The Marconi LID scale? Patients questionnaire? We know that slight dyskinesia may occur early in disease evolution but patients are lot always aware of them and are not always able to recognize them as dyskinesia. It may be, thus, a limit to define the precise onset of LID.

Answer) Because this study focused on the time to onset of dyskinesia, presence or absence of LID is the only data needed. Scales for dyskinesia is for assessment of severity of LID were not used in this study. As reviewer comment, many patients are not aware of their dyskinesia. Therefore, we defined the onset of LID based on clinicians’ observation. But when patients reported onset of dyskinesia and clinician also observed the dyskinesia, we accepted the day that patients indicated. Now we describe this content in Method section of revised manuscript.

- In Table 1 the authors reported the number of patients with treatment other than levodopa: was it the treatment, patients had, at time of LID onset?

Answer) Drug other than levodopa was assessed at the time of LID onset. In revised manuscript, we add this information in Table 1 and Result section.

- In the results section the authors mentioned that female subjects were at higher risk of early LID but they wrote that male subjects were at higher risk in the abstract: please correct.

Answer) We thank for reviewer’s correction. We correct the abstract. 

- The authors associated the risk of LID onset with the rapid rate of levodopa /LED increase. However, it is known that LID are also associated with pulsatile post-synaptic dopaminergic stimulation.

- If data is available, and in order to help interpreting the results of the study, the authors could study and report the way levodopa was prescribed and whether the increase in levodopa dose or LED was associated with an increase in levodopa/LED unitary dose (which may enhance pulsatile answer and promote LID) or whether the increase was associated an increase in the number of daily intakes (a more continuous dopaminergic administration, which may delay the onset of LID). If the authors could add these data, they could develop more precisely the discussion on the pathophysiology of LID and the way to manage PD patients to avoid the occurrence of LID, even in severely depleted patients.

Answer) We selected patients who was treated with levodopa initially, and almost all the patients took levodopa three time a day. And, no patient of this study was treated with continuous intestinal levodopa, subcutaneous apomorphine, rotigotine patch, or deep brain stimulation. Therefore, we think that influence of pulsatile dopaminergic stimulation on onset of LID cannot be explored using this study data and subanalysis suggested by reviewer seems to be impossible in this data

Reviewer #2

Major comments

MATERIALS AND METHODS

Subjects :

- Authors chose to perform the analyses with de novo PD patients. It is well known that some PD patients are in fact misdiagnosed (Coarelli et al, J Neurol Sci, 2019 ; Hustad E, Aasly JO. Front Neurol. 2020). How did they ensure that these patients were truly PD patients and not parkinsonian patients from other degenerative disorders such as PSP, MSA,… Which clinical criteria did they use? Did they check these criteria at the beginning of the « PD » disease but also during the follow-up of these patients? It seems strange that authors did not observe other parkinsonian disorders due to initial misdiagnosis, even if they are experts in movement disorders.

Answer) We did not enroll study patients consecutively, but reviewed medical record and selected proper patients for study. During patient selection, other parkinsonian patients such as atypical parkinsonism, dementia with Lewy bodies, frontotemporal dementia, or unspecified disorders were excluded. Now we describe the process of patient selection in detail in Method section of revised manuscript.

- Did the authors use a score of dyskinesia such as the UDysRS?

Answer) This study focused on the time to onset of dyskinesia, so scales for assessment of severity of LID were not used.

- In the sentence « Data of patients with… drug dose », the « for » seems not appropriate or something is missing.

Answer) According to reviewer’s comment, we revise the sentence.

- How was performed the motor UPDRS by authors? Was it applied in the OFF state or in the ON state (or intermediate state)? How long after the last medication did the UPDRS have been carried out? Please detail the assessment of the UPDRS score.

Answer) UPDRS score rated at de novo status (just before initiation of levodopa) was used in this study. Now we add this content in Method section of revised manuscript.

RESULTS

- Demographic characteristics : it is probably more accurate to rename this part « Demographic and clinical characteristics »

Answer) According to reviewer’s suggestion, we now rename the title of the paragraph and table 1.

- The UPDRS motor score was available for only 89 out of 150 patients. Could you precise how many UPDRS motor scores were available in each group (E, M, L). Moreover, in which condition (On? OFF? intermediate?) were assessed these scores?

Answer) As per reviewer’s suggest, we now present the number of patients of each group who had been assessed with UPDRS motor score and modified Hoehn and Yahr stage (footnote of Table 1). The used score rated at only drug-naïve status.

- The UPDRS is relatively elevated at initiation of Levodopa and/or dopaminergic agonists. It seems that patients were not sufficiently treated. Could you please justify this point?

Answer) First, the UPDRS score was rated at de novo status but not in treatment period. The mean UPDRS motor score at de novo status ranges 20 to 25 in most studies. Moreover, we selected patients who initiated medical treatment with levodopa, so patients with mild symptom who was suitable for dopamine agonist or MAO-B inhibitor monotherapy may be excluded from this study.

- Entacapone increases the occurrence of dyskinesia (Ahn et al., J Clin Neurol. 2007 ; Trenkwalder et al., Neurology. 2019). Did authors observe any difference in dyskinesia occurrence between patients who took entacapone and those who did not take it? Did they look for a difference in each group (E, M, L)?

Answer) The numbers of patients treated with entacapone are already presented in Table 1. There was no difference in number of patients treated with entacapone among E, M, and L groups.

- In the « Risk factor for early LID » section, second line, authors declared that there was a correlation between time to onset of LID and LED but the p value is 0.066 which is above 0.05… (you should also correct the sentence related with this result in the Discussion section)

Answer) As per reviewer’s comment, we revised the sentence for the data.

DISCUSSION

Concerning the sentences from «However the present results… » to « … acts as a trigger for onset » :

It is well established that the occurrence of LID is secondary both to the dopaminergic denervation AND the levodopa administration. Without one of this factor, LID could not appear. LID occur in patients (or animals) taking Levodopa but only in the presence of dopaminergic degeneration.

Answer) In the paragraph, we focused on the role of levodopa dose. In the clinical setting, levodopa dose was affected by severity of dopaminergic denervation, therefore it is unclear whether levodopa dose is an independent factor for LID or higher levodopa dose is just a consequence of severe dopamine depletion. We discussed this point in this paragraph. Now we revise the paragraph to make it less confusing. 

Additionally, as reviewer commented, both dopaminergic denervation and administration of dopaminergic drug are essential for onset of LID. But, Tagasaki et al. (Neuropharmacology 2005) showed that therapeutic dose of levodopa induced LID in DAT blocker-treated squirrel monkey without dopaminergic denervation, and Pearce et al. (Psycholpharmacology 2001) demonstrated that higher dose of levodopa induced LID in dose dependent manner in normal macaque monkey. Therefore, we think that the role of levodopa on LID should be reconsidered. 

Minor comments

- Replace « Table 2 » by « Table 1 » in the « Drug » section of the Results.

- Insert « Table 2 » at the end of the sentence « Multivariable linear regression models … were associated with early onset of LID ». Same remark for the next sentence.

Answer) We appreciate reviewer’s correction. Now we edit that according to the comment.

---

## [Decision Letter · Decision Letter 1]

15 Jul 2020

PONE-D-20-13495R1

Rapid drug increase and early onset of levodopa-induced dyskinesia in Parkinson's disease

PLOS ONE

Dear Dr. Seo Hyun Kim,

Thank you for submitting your manuscript to PLOS ONE. After careful consideration, we feel that it has merit but does not fully meet PLOS ONE’s publication criteria as it currently stands. Therefore, we invite you to submit a revised version of the manuscript that addresses the points raised during the review process.

Please, answer to the remaining minor comments adressed by the referees.

We look forward to receiving your revised manuscript.

Kind regards,

Véronique Sgambato

Academic Editor

PLOS ONE

Reviewers' comments:

Reviewer's Responses to Questions

**Comments to the Author**

1. If the authors have adequately addressed your comments raised in a previous round of review and you feel that this manuscript is now acceptable for publication, you may indicate that here to bypass the “Comments to the Author” section, enter your conflict of interest statement in the “Confidential to Editor” section, and submit your "Accept" recommendation.

Reviewer #1: (No Response)

Reviewer #2: All comments have been addressed

2. Is the manuscript technically sound, and do the data support the conclusions?

Reviewer #1: Yes

Reviewer #2: Yes

3. Has the statistical analysis been performed appropriately and rigorously? 

Reviewer #1: Yes

Reviewer #2: Yes

4. Have the authors made all data underlying the findings in their manuscript fully available?

Reviewer #1: Yes

Reviewer #2: Yes

5. Is the manuscript presented in an intelligible fashion and written in standard English?

Reviewer #1: Yes

Reviewer #2: Yes

6. Review Comments to the Author

Reviewer #1: Rapid drug increase and early onset of levodopa-induced dyskinesia in Parkinson's disease

PONE-D-20-13495 R1

Research article for the journal Plos One

The authors have revised the manuscript of a retrospective multicenter study on the occurrence of levodopa induced dyskinesia (LID) in Parkinson Disease (PD) patients.

They answered to all comments of the reviewers and corrected the errors in the previous version of the manuscript.

They added data if available.

They have developed more precisely the discussion section.

The manuscript is now clearer than the first version.

The authors did not add any comment on pharmacokinetics and the role of pulsatile administration of levodopa in sensitization and in the development of motor complications (see Nutt J, 1995, Clin Exp Pharmacol Physiol, Nutt J, 2007, Movement Disorders Journal, Sharma et al, 2015 Biomed Pharmacother for example). Most of the patients were taking treatment 3 times a day, as noted by the authors in the answers to comments. It did not seem to be different across groups. This would argue against the idea that high unitary doses of levodopa increase the risk of LID, whereas the rate of increase seem to be more important.

They could perhaps add this comment in the limits of the study section.

They could also add their answer on reviewer 2 comment on the discussion section.

Even though the authors could not confirm that the risk to develop LID is associated with the rate of levodopa increase rather than the initial dose or the severity of dopamine denervation, these data are interesting as they provide clues for the management of patients when trying to avoid LID.

Reviewer #2: Authors responded to all the queries.

I just have few minor remarks :

- Please use « levodopa » or « Levodopa » but authors should homogeneize this in all the manuscript

- Lines 74-75, replace « Among them, we selected patients who medical treatment initiated with levodopa » by « Among them, we selected patients who initiated medical treatment with levodopa »

- Lines 82-84 : I think authors should rewrite the sentence « When patient reported symptom suggesting LID or dyskinetic movement was observed in the clinic, clinicians have been recording the information for LID » ; « When patient reported symptom suggesting LID or dyskinetic movement was observed in the clinic, clinicians recorded the information for LID

- Line 89 : « developed within 2 year … » ; add a « s » for years : « developed within 2 years … »

- Line 94 : please put « de novo » in italic

- Table 1 : « P-value » should be replaced by « p-value »

7. PLOS authors have the option to publish the peer review history of their article (what does this mean?). If published, this will include your full peer review and any attached files.

Reviewer #1: No

Reviewer #2: No

---

## [Author Response · Author response to Decision Letter 1]

23 Jul 2020

Reviewer #1

The authors did not add any comment on pharmacokinetics and the role of pulsatile administration of levodopa in sensitization and in the development of motor complications (see Nutt J, 1995, Clin Exp Pharmacol Physiol, Nutt J, 2007, Movement Disorders Journal, Sharma et al, 2015 Biomed Pharmacother for example). Most of the patients were taking treatment 3 times a day, as noted by the authors in the answers to comments. It did not seem to be different across groups. This would argue against the idea that high unitary doses of levodopa increase the risk of LID, whereas the rate of increase seem to be more important.

They could perhaps add this comment in the limits of the study section.

Answer) Now, we add the contents about the way levodopa was initiated and increased in Method section (lines 104-106 in revised manuscript). In this study, levodopa was usually prescribed three times a day with equal dose, and we increased the levodopa by 50mg per dose (150mg a day) when dose increased was needed. As we answered in previous response to review, there was no change in the number of daily intakes of levodopa during dose increase and there was no difference in number of daily intakes among patients. Therefore, we think that issue for the continuous dopaminergic stimulation is not applicable to this study.

Reviewer #2

- Please use « levodopa » or « Levodopa » but authors should homogeneize this in all the manuscript

Answer) Now we use “levodopa” in this revision.

- Lines 74-75, replace « Among them, we selected patients who medical treatment initiated with levodopa » by « Among them, we selected patients who initiated medical treatment with levodopa »

- Lines 82-84 : I think authors should rewrite the sentence « When patient reported symptom suggesting LID or dyskinetic movement was observed in the clinic, clinicians have been recording the information for LID » ; « When patient reported symptom suggesting LID or dyskinetic movement was observed in the clinic, clinicians recorded the information for LID

Answer) We revise the sentences according to the suggestion.

- Line 89 : « developed within 2 year … » ; add a « s » for years : « developed within 2 years … »

- Line 94 : please put « de novo » in italic

- Table 1 : « P-value » should be replaced by « p-value »

Answer) We correct the errors.

---

## [Editor Report · Decision Letter 2]

28 Jul 2020

Rapid drug increase and early onset of levodopa-induced dyskinesia in Parkinson's disease

PONE-D-20-13495R2

Dear Dr. Seo Hyun Kim,

We’re pleased to inform you that your manuscript has been judged scientifically suitable for publication and will be formally accepted for publication once it meets all outstanding technical requirements.

Kind regards,

Véronique Sgambato

Academic Editor

PLOS ONE
---

## [Editor Report · Acceptance letter]

10 Aug 2020

PONE-D-20-13495R2 

Rapid drug increase and early onset of levodopa-induced dyskinesia in Parkinson's disease 

Dear Dr. Kim:

I'm pleased to inform you that your manuscript has been deemed suitable for publication in PLOS ONE. Congratulations! Your manuscript is now with our production department. 

Kind regards, 

on behalf of

Dr. Véronique Sgambato 

Academic Editor

PLOS ONE